# Collagen Obtained from Leather Production Waste Provides Suitable Gels for Biomedical Applications

**DOI:** 10.3390/polym14214749

**Published:** 2022-11-05

**Authors:** Lesia Maistrenko, Olga Iungin, Polina Pikus, Ianina Pokholenko, Oksana Gorbatiuk, Olena Moshynets, Olena Okhmat, Tetiana Kolesnyk, Geert Potters, Olena Mokrousova

**Affiliations:** 1Department of Biotechnology, Leather and Fur, Kyiv National University of Technologies and Design, 01011 Kyiv, Ukraine; 2Institute of Molecular Biology and Genetics of the National Academy of Sciences of Ukraine, 03143 Kyiv, Ukraine; 3Antwerp Maritime Academy, 2030 Antwerp, Belgium; 4Department Bioscience Engineering, University of Antwerp, 2020 Antwerp, Belgium

**Keywords:** collagen, collagen extraction, leather production wastes, cell culture

## Abstract

Collagen and its derivates are typically obtained by extracting them from fresh animal tissues. Lately, however, there has been an increased interest in obtaining collagen from other sources, such as waste material, because of the growing trend to replace synthetic materials with sustainable, natural counterparts in various industries, as well as to ensure a rational waste revalorization. In this paper, collagen was obtained from non-tanned waste of leather production, taken at different stages of the production process: limed pelt, delimed pelt, and fleshings. A stepwise extraction through acid hydrolysis in 0.5 M acetic acid and subsequent precipitation with NaCl lead to collagen-containing protein extracts. The highest collagen yield was achieved in extracts based on delimed pelt (2.3% m/m after a first extraction round, and an additional 1.4% m/m after the second round). Hyp/Hyl molar ratios of 10.91 in these extracts suggest the presence of type I collagen. Moreover, gels based on these collagen extracts promote adhesion and spreading of HEK293 cells, with cells grown on collagen from delimed pelt showing a larger nuclear and cell expansion than cells grown on traditional bovine tendon atelocollagen. This suggests that these collagen gels are promising natural biomedical carriers and could be used in a wide range of medical and cosmetic applications.

## 1. Introduction

Collagens are proteins which form the basis of connective tissue and represents about 30% of total protein mass of the body in mammals [1]. It occurs almost in all tissues both in animals and in humans, and is widely found in skin, bones, blood vessels, cartilage, dentin of teeth, etc. More than 29 different types of collagens have been reported, consisting of 46 different chains of polypeptides, which are classified by structure into fibril-forming, network forming, fibril-associated collagens with interrupted triple helices (FACIT), membrane-associated collagens with interrupted triple helices (MACIT), and multiple triple-helix domains and interruptions (MULTIPLEXINs) [2,3]. All types of collagens have a characteristic triple helix, but the length of the helix, the size, and nature of the non-screw part vary depending on the type. Most collagen for industrial applications is currently obtained from mammalian skin, particularly from cattle, with the main and most abundant types of collagens present in the skin are types I and III [4]. However, collagen can also be extracted from alternative, marine-based resources such as fish, octopus, starfish, sea-cucumber, and jellyfish [5,6,7]. In fact, different tissues are used for the extraction of different types of collagens. For example, the Achilles tendon is used to produce both collagen type I and III, placental villi are a preferential source of type IV, and bovine nasal or articular cartilage are used for getting collagen type II.

Mammalian collagen has high therapeutic potential in pharmaceutical, biomedical, and cosmetic industries [8]. Due to the unique amino acid composition of this protein, which is highly conserved among mammals, collagen demonstrates a low immunogenicity, which in turn contributes to a wide range of medical and biotechnological application. For example, cattle skin collagen is used to strengthen tendons and wound healing (using a collagen matrix); neonatal bovine dermis is used for hernia recovery, plastic, and reconstructive surgery; and the pericardium of adult cattle is applied to strengthen muscles during recovery [9,10]. The use of such collagen is limited due to the high production cost, with direct collagen extraction from animal tissues believed to be the most cost-effective method for now. One of the possible ways to make it more cost-effective is to find an adequate and cheap source of collagen.

Luckily, there is still an untapped source of collagen. The leather industry produces significant amounts of liquid and solid waste: depending on the skin type, no more than 20% of the raw skin material (by weight) is used for further leather production. Also, the remaining waste may even contain chromium. Up to 850 kg of solid waste is generated from 1000 kg of raw materials during leather processing, with 80% of this waste non-tanned, and 20% tanned [11]. Moreover, instead of recycling this waste material to obtain valuable products (proteins, amino acid and chromium concentrates, etc.), most of it is taken out to landfills causing environmental pollution. These wastes contain up to 50% of economically valuable protein. For example, dry dermis consists for about 70–80% of fibrous collagen as main protein, apart from other commercially important proteins such as keratin, elastin, reticulin, albumin, as well as proteoglycans [12]. With the help of appropriate recycling technologies, these valuable proteins can be extracted for further, wide use, in collagen-containing applications in various industries. Upcycling collagen-containing wastes from leather production may therefore bring environmental, health-promoting, and economic advantages and may as such contribute to the United Nations Sustainable Development Goals.

The aim of this study was therefore to estimate the feasibility to isolate collagen from collagen-containing wastes derived from a leather production process, and to demonstrate that these collagen extracts can be used in pharmaceutical and biomedical applications.

## 2. Materials and Methods

### 2.1. Materials

Limed pelt, delimed pelt, and fleshings were chosen as different types of waste and were obtained from the cattle tannery “Chinbar” (Kyiv, Ukraine).

### 2.2. Collagen Extraction

Collagen was obtained through a basic method of acid hydrolysis [13,14]. All types of waste have been subjected to three rounds of extraction (1st extraction, 2nd extraction and 3rd extraction) according to the protocols presented in Table 1.

The collagen yield was determined by combining the measurements of the total nitrogen content and of the amounts of protein in the collagen solutions after extraction and subsequent dissolution in acetic acid. Collagen was re-extracted twice for all studied waste types, from the remaining material after the first and second stages of extraction, to determine the economic expediency of the second and third stages of extraction. 

### 2.3. The Content of Total Nitrogen

Total nitrogen content was determined by the Kjeldahl method [15,16]. An exact portion of the test sample and 1 g of a ground mixture of potassium sulphate and copper sulphate (ratio of 10:1) were put in a 200–300 mL Kjeldahl flask, to which 7 mL of concentrated sulfuric acid was added. The flask was gradually heated, closed with a glass funnel, and boiled on an electric heater for several hours to obtain a light green solution. The flask was then boiled for at least another 30 min until the solution became clear. The flask was cooled by carefully adding 20 mL of water and then connected to the assembled nitrogen tester. In the receiver, before distillation, 20 mL of 4% boric acid and 0.25 mL of indicator were added. The lower end of the inner tube of the refrigerator was thereby lowered into the solution in the receiver. Subsequently, 40 mL of 30% sodium hydroxide were slowly added to the funnel flask. About 100 mL of distillate was collected. The heating was stopped and the Kjeldahl flask was immediately disconnected from the appliance. After distillation, the distillate was titrated with 0.1 M hydrochloric acid until the colour of the mixed indicator changed. A control experiment was treated in the same way and with the same reagents, but without the test sample and the result thereof was used to correct the calculation of the nitrogen content. Determination of total nitrogen content was performed on the basis that 1 mL of hydrochloric acid corresponds to 1.401 mg of total nitrogen.

### 2.4. Determination of Moisture Content

The moisture content of the samples was determined by weight by drying a portion of experimental collagen-containing waste in an oven at a temperature of 170 °C. A portion of the crushed samples weighing 4 g was weighed on an analytical balance, and placed in an oven for 1 h. The box was cooled in a desiccator to a temperature of 20 °C and weighed again on an analytical balance. Drying, cooling, and weighing of the box was repeated to constant weight, until the difference between subsequent weighings did not exceed 0.001 g. The allowable difference for parallel experiments was not more than ±0.3%.

### 2.5. Mineral Content

Determination of mineral content was performed by the weight method by burning a sample of experimental collagen-containing wastes in a muffle furnace at a temperature of 500–600 °C, according to ISO 4047:1977|IULTCS/IUC7, as a standard for industrial leather analysis. An aliquot of 4 g of the crushed samples was weighed on an analytical balance and placed in a muffle furnace. Burning was considered complete when the colour of the sample was grey without black specks. The sample was subsequently cooled in a desiccator to a temperature of 20 °C and weighed on an analytical balance. Calcination, cooling and weighing of the sample were repeated to constant weight when the difference between adjacent weighings did not exceed 0.001 g. The permissible difference for parallel experiments was not more than ±0.1%.

### 2.6. Calcium Hydroxide Content

To determine the calcium hydroxide content, the sample, after the determination of its mineral content, was quantitatively transferred to a heat-resistant flask using 100 mL of distilled water and boiled for 5 min. After cooling, the contents of the flask were titrated with 0.01 M hydrochloric acid solution in the presence of a phenolphthalein indicator. Determination of calcium hydroxide content was performed on the basis that 1 mL of 0.01 M hydrochloric acid corresponds to 0.37 mg of calcium hydroxide.

### 2.7. Measurement of the Total Protein Content by the Biuret Method

A specific reaction with proteins can be observed by the biuret reaction [17] as it indicates the presence of polypeptide links. All proteins, peptones, and polypeptides, from tetrapeptides onwards, can take part in a biuret reaction. The colour intensity is proportional to the number of peptide links, and consequently to the concentration of protein in solution. The main test was made as follows: 0.1 mL of test solution with 5.0 mL of biuret reagent was added to a test tube and mixed, with a sample of a 0.9% sodium chloride solution as a control. After 30 min, the optical density was checked with a Spectrophotometer ULAB 102 (China) at 540 nm.

### 2.8. Measurement of the Total Protein Content by the Bradford Method

The Bradford method [18] is characterized by a high sensitivity (down to 0.2 μg), as well as a relatively high speed of analysis and a good reproducibility of results. The method allows to determine the protein content in samples with an accuracy of 0.5–50 μg/mL and can be applied to proteins dissolved in electrophoretic buffer containing mercaptoethanol and Tris-HCl. The method is based on the formation of coloured Coomassie complexes of diamond blue G-250 with macromolecules of polypeptides due to electrostatic interactions. The maximum absorption of the complex in the visible region does not coincide with the maximum absorption for pure dye, which allows quantitative measurement of the complexes that are formed. To 0.2 mL of diluted protein extract, 2 mL of dye solution was added, followed by gentle mixing. After 2 min, the optical density was measured at a wavelength of 595 nm against the control (dye solution with buffer used to obtain the extract in appropriate volumes). Measurements were performed in a cuvette with a light-absorbing layer thickness of 10 mm on Spectrophotometer ULAB 102 (Shanghai, China).

### 2.9. Ion Exchange Liquid Column Chromatography of Amino Acids

Ion exchange liquid column chromatography was performed according to the methodology of [19]. Sample preparation is the first prerequisite for obtaining reliable and reproducible results when an automatic amino acid analyser is used. The sample preparation process can be divided into the isolation of amino acids bound in proteins, peptides that require hydrolysis, and the preparation of samples containing free amino acids (biological fluids, tissue extracts), from which proteins and other substances that interfere with analysis are removed. Hydrochloric acid was used for hydrolysis: a carefully weighed sample with a dry protein content of 2 mg or an equivalent amount of aqueous protein solution was placed at the bottom of the test tube made of refractory glass. As a control, 0.5 mL of distilled water and 0.5 mL of concentrated hydrochloric acid were added to a test tube. The same amount of concentrated hydrochloric acid was added to each of the aqueous protein solutions. The tube was cooled in a mixture of dry ice with acetone/liquid nitrogen. After the samples were frozen, air was pumped out with vacuum pump to prevent the oxidation of amino acids because of hydrolysis. Afterwards, the test tubes were sealed and placed for 24 h in a thermostat (106 °C). At the end of the hydrolysis, the tubes were opened, pre-cooled to room temperature, and the samples were transferred to a glass box and placed in a vacuum desiccator over granular caustic soda. The air was removed from the desiccator using a water jet pump. After drying the sample, 3–4 mL of deionized water was added to the beaker and the drying procedure was repeated. The samples prepared in this way were dissolved in a 0.3 N lithium citrate buffer (at pH 2.2) and were applied onto the ion exchange column of the amino acid analyser. Deproteinization (protein precipitation) of samples to obtain an extract of free amino acids and low molecular weight compounds (peptides) was performed as follows: 1 mL of the test protein solution was placed in a clean centrifuge tube, 1 mL of a 3% aqueous solution of sulfosalicylic acid was added, and the result was mixed thoroughly. After that, the precipitated protein was separated by centrifugation at 3500–4500 rpm for 30 min. The supernatant was applied onto the ion exchange column of the amino acid analyser and the procedure described above was performed. Determination of the amino acid composition of the studied collagen solutions was performed on an automated amino acid analyser T339 (Mikrotehna, Czech Republic).

### 2.10. Cell Adhesion on Different Collagen Coatings

The HEK293 (human embryonal kidney) cell line was grown as a monolayer in DMEM/high glucose (Biowest, Nuaillé, France) supplemented with 10% of foetal bovine serum (Sigma-Aldrich, St. Louis, MO, USA), penicillin 100 U/mL (Arterium, Kyiv, Ukraine), and streptomycin 100 µg/mL (Arterium, Ukraine). After the cells reached 70–80% confluence, they were passaged by standard method using a trypsin–EDTA mixture. Cell adhesion tests were performed on collagen coatings obtained via the different extraction procedures.

As a positive control, plastic with adsorbed telopeptides of type I bovine collagen was used. Additionally, an uncoated surface which was pretreated with a 0.5 M solution of acetic acid and washed twice with phosphate–salt buffer at pH 7.4, served as a negative control. 2 × 10^5^ HEK293 cells were seeded on a 35 mm diameter Petri dish in 1.5 mL of serum-free DMEM/high glucose nutrient medium. Cells were incubated at 37 °C and 5% CO_2_ for 60 min. After that, the culture medium and non-adherent cells were removed, and the surface was washed three times with phosphate–salt buffer at a pH of 7.4. Cells were fixed and stained according to Pappenheim [20].

To determine the morphometric parameters of the cells after adhesion, the samples were stained according to Pappenheim and analysed microscopically at 70 cells in 10 fields of view per each group. Each field was obtained at a magnification of 20 × 10 (microscope Leica and digital camera Sigeta). The morphometric parameters were determined using the ImageJ software (version 1.52a).

The following morphometric parameters were calculated: the average area of the HEK293 cells and their nuclei (Sn), both expressed in µm^2^. These values were used to calculate the area of the cytoplasm (Sc) in µm^2^, as the difference between the total cell area and the area of the nuclei, as well as the Nuclear–Cytoplasmic Ratio (NCR), as the ratio of the area of the nucleus to the area of the cytoplasm (NCR = Sn/Sc) [21].

### 2.11. Statistical Data Analysis

All measurements were done in three parallel replicas. Comparisons involving multiple groups were performed using a one-way analysis of variance (ANOVA) with a Tukey post hoc test [22]. For all tests, *p* < 0.05 was considered significant. Data analysis was carried out using the Prism package (GraphPad Software, San Diego, CA, USA).

## 3. Results and Discussion

### 3.1. General Composition of the Starting Material

The main components in the waste samples were moisture, minerals, total nitrogen, and calcium hydroxide (Table 2).

The increased content of minerals and calcium hydroxide in the limed pelt and limed fleshing samples can be explained by the technological process of leather production, which involves the processing of raw materials in a calcium hydroxide concentrated solution. As the deliming process serves mainly to extract this calcium hydroxide from the dermis, the content of both minerals and calcium hydroxide in the delimed pelt after trimming was 3.2–9.6 and 6.5–16.8 times lower, respectively. The lowest content of total nitrogen, which indirectly indicates the amount of collagen in the structure, was found in the limed fleshing samples.

### 3.2. Stages of Collagen Extraction from Wastes Samples

The main criterion for the evaluation of the efficiency of the collagen extraction was the amount of extracted collagen. The amount of collagen obtained after the first and subsequent stages of extraction from different types of leather waste have been provided (Table 3). The largest amount of total nitrogen was detected after the second extraction. The amount of total nitrogen was 45% less in limed pelt and 84% less in fleshings compared to delimed pelt.

The largest amount of protein (determined with the biuret method) was detected after the first extraction for all experimental groups. Moreover, the highest protein amount was observed in delimed pelt, which was 2.5 and 8.4 times higher compared to limed pelt and fleshings, respectively. The high content of total nitrogen in the solutions after the second extraction compared to the protein amount could be the evidence of additional leaching of non-protein structures that contain nitrogen or excessive destruction of amino acids by acetic acid solution.

The smaller amount of protein in limed pelt and fleshing samples could be explained by the fact that part of the acetic acid from the solution was spent on neutralizing the excess of calcium hydroxide found in these samples. The fleshing samples were also characterized by a lower initial content of total nitrogen (Table 1). To sum up, the largest amount of collagen with the highest yield of total nitrogen was obtained from the samples of delimed pelt. The third extraction step was not efficient due to the low amount of extracted collagen for each waste type. Therefore, the Bradford protein assay was performed only for collagen solutions obtained after the first and second extraction (groups I1, I2, II1, II2) (Table 4). Fleshing samples were not tested due to the small amount obtained after the third stage of extraction.

A determination of the protein concentration in the experimental solutions by the biuret method and the Bradford method show almost identical data. Solutions with similar concentrations of protein can be assumed to produce enough collagen further downstream.

### 3.3. Identification and Amino Acid Composition of the Collagen Extracts

The identity of the protein being collagen was shown previously by SDS-PAGE [23]. However, there are additional ways to identify the protein in the extracts, such as by amino acid composition. The amino acid composition of collagen is extremely specific and differs sharply from the amino acid composition of other proteins. Collagen contains specific marker amino acids such as hydroxyproline and hydroxylysine (up to 23%), whereas there is no tryptophan and very little cysteine, tyrosine, and methionine in collagen samples. The full amino acid composition of the studied collagen solutions is provided in Table 5, expressed as mg of each of the amino acids per 100 mL of solution (ratio test solution-buffer was 1:20).

Various collagen types significantly differ in the Hyp/Hyl molar ratio, and this parameter can be used for their identification. The Hyp/Hyl molar ratio was calculated for each sample to describe collagen type (Table 6). The result for limed pelt after the first extraction and delimed pelt after the second extraction suggests the presence of type I collagen [24]. All other samples show a rather low ratio which makes it difficult to identify the collagen type.

### 3.4. Collagen Extraction Efficiency and Yield Estimates

Since hydroxyproline is a kind of amino acid marker of collagen and elastin, a comparative analysis for its content was performed in the experimental solutions. The highest concentration of hydroxyproline was found in the solutions after the first extraction (Figure 1), as well as in the solution from limed pelt (1.1 and 5.3 times more compared to delimed pelt and fleshings samples, respectively). However, based on the volume of collagen solutions obtained after the first and second extractions, the highest yield of hydroxyproline by weight (and hence collagen) was observed in the samples obtained from delimed pelt (Figure 1).

The amount of waste accumulated in tanneries depends on the particular source of the material. However, the mean amounts have been presented in Table 7 for each by-product. Based the extraction protocols (Table 1) and given the extraction efficiencies calculated in Figure 1, it is possible to obtain about 685 L of collagen-containing solutions from 1 ton of delimed pelt waste and about 257 L from limed pelt waste. In any case, the relation between the measurements of protein content (Table 4) as well as amino acid composition analysis (Table 5) can be used as markers to identify the extracts (in this case, I and IV) from which sufficient collagen can be obtained.

### 3.5. Collagen Gel Formation

The final stage of sample preparation occurred through a buffer exchange in collagen solutions from acetic acid to a number of solutions (deionized water, 18 MOm·cm, pH 7; 0.1 M NaOH, pH 13; 10% aqueous NaCl solution, pH 7) and subsequent testing for gelation by dialysis of these solutions at 4 °C for 12 h (Figure 2). The best gelation was observed using deionized water. Strong clear gels were obtained from collagen samples from limed and delimed pelt. The collagen solution from fleshing samples did not gel after dialysis, which can be explained by the low concentration of protein in the solution. As an alternative to acetic acid, a 0.5 M citric acid solution was used to resolubilize the collagen for further gelation experiments. However, while solutions in 0.5 M acetic acid could safely be kept for at least three years, solutions in 0.5 M citric acid showed the development of fungi in the solution after no more than two weeks.

### 3.6. Cell Adhesion

To demonstrate the possible applicability of the collagen extracts in biomedical applications, the adhesion of HEK293 cells was compared on uncoated tissue-culture grade polystyrene plastic, on collagen-coated polystyrene with collagen obtained from limed and delimed pelts, and on polystyrene coated with bovine tendon atelocollagen (with the latter being tested as a positive control). This cell line is among the most frequently used mammalian cell lines for a wide range of applications including recombinant protein production, manufacturing of viral vectors, cancer research, and cytotoxicity and biocompatibility studies [25,26]. HEK293 cells adhered to the uncoated and collagen-coated surfaces, but the cell morphology was affected by the presence of the coating (Figure 3).

Morphometric data analysis showed that the average area of the nuclei of the HEK293 cell line, grown on gels with collagen derived from delimed pelt, was significantly larger than in the positive control group (Figure 4A). At the same time, the area of the HEK293 cells significantly increased when grown on collagen from delimed pelt compared to the positive control group (Figure 4B). In this case, cell adhesion is associated with cell flattening due to an increase of the cytoplasm area. Therefore, in general, cell adhesion correlates with the NCR (Figure 4C). Moreover, the NCR parameter obtained from the collagen from the limed pelt group had significantly lower values than the NCR of the positive control group, and the lower the ratio, the greater the cell adhesion.

Collagen isolated from different types of raw materials can therefore stimulate the adherence and spreading of mammalian cells. It is known that the attachment of cells to collagen occurs due to its interaction with a number of cell receptors, such as receptors of the integrin family (α1β1, α2β1, α10β1, and α11β1), as well as GPVI, DDR1 and DDR2, LAIR-1, and others [27]. Since cell spreading was noted to accelerate, which is mediated by the interaction of integrin receptors on the surface of the substrate, it can be assumed that in the process of obtaining collagen derivatives, the RGD sequences responsible for interaction with integrin receptors were at least partially preserved in their native, unmodified state. In any case, collagen obtained from waste products (limed and delimed pelt) promotes the attachment of HEK293 cells in vitro, which suggests the promising use of this waste-derived collagen in further biomedical applications.

### 3.7. Advantages and Limitations of the Proposed Extraction Method

The data presented here demonstrate that it is in principle possible to extract high quality collagen from low-cost materials, i.e., industrial waste from the leather industry, and that this collagen offers at least the option to be used in high-level, high-cost applications downstream. It should nevertheless be clear that the protocols presented here need further refining, as a yield of only 2% by mass (Figure 1) should still be significantly increased before industrial processing can be considered viable. Moreover, with extraction efficiencies like that, it is still highly questionable if other proteins, which are there in much lower quantities, can be extracted in sufficient quantities for specific applications.

Secondly, no attention has been directed towards the possibility to extract collagen from leather waste under sterile conditions as needed for further biomedical applications downstream. While the obtained collagen could be stored for three years in acetic acid, gels containing this collagen showed signs of fungal growth already after one month. The alternative will be to set up a methodology to sterilize the material after extraction.

Lastly, not all leather wastes have been tested with this protocol, as the tests presented here only used waste from cattle leather production and not from pig or sheep skin. Also, the aforementioned wastes containing chromium (produced after leather tanning) are obviously unsuitable to deliver materials that can be put into contact with living cells, unless the removal of the chromium ions can be assured. In terms of purity, the gels created with the material presented here show no cytotoxicity (towards the HEK293 cells), but this may change once other sources or polluted wastes are used.

## 4. Conclusions

Non-tanned wastes of leather production (limed pelt, delimed pelt) could be a promising source of collagen for biomedical applications. Two consequent extractions of delimed pelt provide the highest protein yield and content of hydroxyproline (amino acid marker of collagen). The third extraction was not efficient due to the low amount of extracted collagen for each wastes type. Strong transparent collagen hydrogels obtained from samples of limed and delimed pelt promoted HEK293 cells surface adhesion and enhanced spreading, which both demonstrate the biocompatibility of the collagen samples obtained with the method described here.

## Figures and Tables

**Figure 1 polymers-14-04749-f001:**
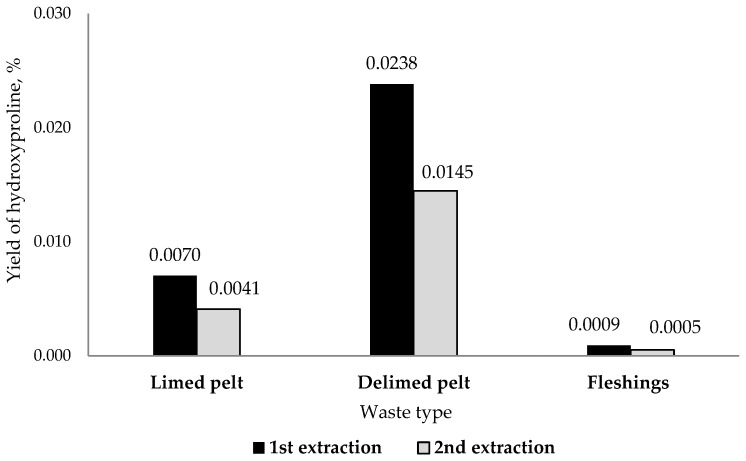
Hydroxyproline yield from different types of waste, expressed as a % (mass ratio) of the initial waste mass.

**Figure 2 polymers-14-04749-f002:**
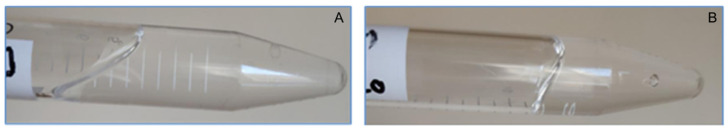
Collagen gels obtained after dialysis: (**A**)—limed pelt (group I), (**B**)—delimed pelt (group IV).

**Figure 3 polymers-14-04749-f003:**
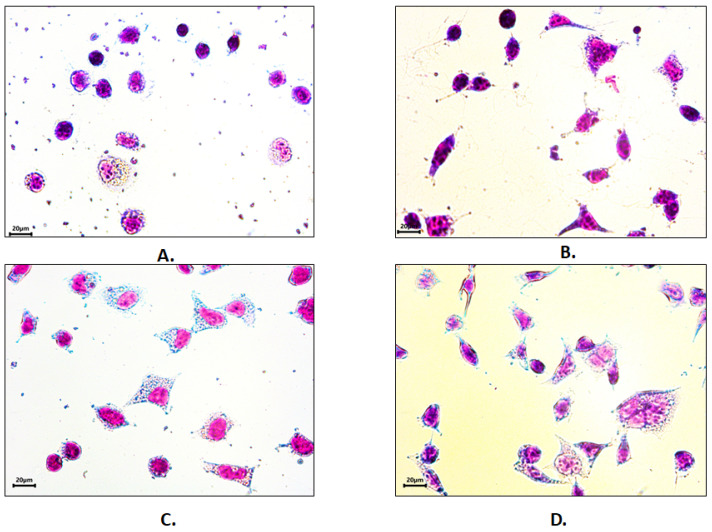
HEK293 cells adhesion and morphology on (**A**) synthetic (polystyrene) and biomimetic (**B**) bovine atelocollagen, (**C**) bovine collagen (limed pelt), and (**D**) bovine collagen (delimed pelt) treated surfaces. Pappenheim staining. Bar 20 µm.

**Figure 4 polymers-14-04749-f004:**
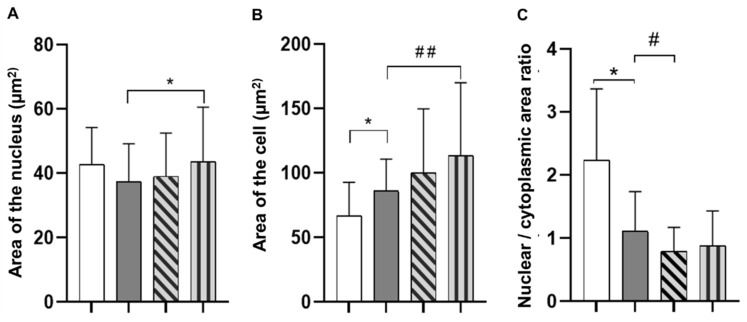
Morphometric parameters of the HEK293 cell line. (**A**) Nuclear area (Sn). *p* values were calculated by ANOVA with Tukey’s test. * *p* < 0.05—positive control vs. collagen from delimed pelt. (**B**) Cell area. *p* values were calculated by ANOVA with Tukey’s test. * *p* < 0.05—negative control vs. positive control, ## *p* < 0.05—positive control vs. collagen from delimed pelt. (**C**) *p* values were calculated by ANOVA with Tukey’s test. * *p* < 0.05—negative control vs. positive control, # *p* < 0.05—positive control vs. collagen from limed pelt. For all graphs: white bar: negative control; grey bar: positive control; diagonally striped bar: collagen from limed pelt; vertically striped bar: collagen from delimed pelt.

**Table 1 polymers-14-04749-t001:** Collagen extraction protocol.

Step	Stage	Extraction		
		1st	2nd	3rd
1	Deliming with ammonium sulphate 3% by weight of the samples, duration 1 h, temperature 38–40 ° C (only to obtain samples of delimed pelt)	+	–	–
2	Grinding samples to a size of 3 × 3 mm	+	–	–
3	Weighing	+	–	–
4	Rinsing in water at a temperature of 20 °C for 45 min with water change each 15 min	+	–	–
5	Extraction of non-collagen proteins with a 10% sodium chloride solution, 1 h of rotation on a shaker at 20 °C, then 22 h of rest at 4 °C and again 1 h of rotation on a shaker at 20 °C	+	–	–
6	Rinsing of the samples with distilled water at pH = 6.5	+	–	–
7	Extraction of collagen with 0.5 M acetic acid solution in the presence of 5 mm EDTA in a ratio of 1:10 (weight: volume), 2 h of rotation on a shaker at 20 °C, then 20 h of rest at 4 °C and again 2 h of rotation at 20 °C	+	+ *	+ **
8	Filtering through a paper filter. The crushed samples were used for further extraction steps.	+	+	+
9	Precipitation of collagen from the filtrate with dry sodium chloride for 24 h at 4 °C	+	+	+
10	Centrifugation for 30 min at 3000 rpm	+	+	+
11	Dissolution of precipitated collagen in minimal volume of 0.5 M acetic acid	+	+	+
12	Re-precipitation of collagen with dry sodium chloride (to a concentration of the latter in a solution of 0.9 M) for 24 h at 4 °C	+	+	+
13	Centrifugation for 30 min at 3000 rpm	+	+	+
14	Dissolution of precipitated collagen in minimal volume of 0.1 M acetic acid	+	+	+

* after the first extraction, ** after the second extraction.

**Table 2 polymers-14-04749-t002:** Chemical composition of leather production wastes.

Mass Fraction, %				
Waste Type	Moisture	Minerals *	Total Nitrogen *	Calcium Hydroxide *
Limed pelt	82.1 ± 0.2	10.7 ± 0.2	14.3 ± 0.4	2.6 ± 0.2
Delimed pelt	80.4 ± 0.2	3.3 ± 0.2	15.0 ± 0.4	0.4 ± 0.2
Fleshings	84.2 ± 0.2	31.7 ± 0.2	5.4 ± 0.4	6.7 ± 0.2

* Measurements were performed on absolutely dry matter.

**Table 3 polymers-14-04749-t003:** The amount of collagen after the 1st, 2nd, and 3d extractions.

Group	Waste Type	Extraction	Volume of Collagen Solution, mL	Total Nitrogen, mg	Protein Amount (Biuret Method), mg
I	Limed pelt	1st	18.0 ± 0.90	7.6 ± 0.04	10,1 ± 0.10
II		2nd	70.0 ± 3.50	39.2 ± 0.20	5.0 ± 0.05
III		3rd	16.0 ± 0.80	3.3 ± 0.02	2.2 ± 0.02
IV	Delimed pelt	1st	48.0 ± 2.40	50.4 ± 0.25	25.3 ± 0.03
V		2nd	85.0 ± 4.25	71.4 ± 0.36	11.4 ± 0.11
VI		3rd	15.0 ± 0.75	1.0 ± 0.01	4.7 ± 0.05
VII	Fleshings	1st	12.5 ± 0.63	3.3 ± 0.02	3.0 ± 0.03
VIII		2nd	5.0 ± 0.25	11.5 ± 0.06	1.4 ± 0.01
IX		3rd	2.5 ± 0.13	0.5 ± 0.01	0.6 ± 0.01

**Table 4 polymers-14-04749-t004:** Protein concentration in studied samples.

Protein Concentration (mg/100 mL)		
Group	Biuret Method	Bradford Method
I	56.11 ± 0.56	66.97 ± 0.67
II	7.14 ± 0.07	13.66 ± 0.14
IV	52.71 ± 0.53	63.51 ± 0.64
V	13.41 ± 0.13	34.23 ± 0.34

**Table 5 polymers-14-04749-t005:** Amino acid composition of extracts.

Amino Acid	Amount, mg					
	I	II	IV	V	VII	VIII
Hydroxylysine	3.214	1.350	3.780	1.350	1.746	1.350
Lysine	12.746	10.950	19.467	12.167	11.536	12.167
Histidine	2.627	1.788	4.769	3.726	3.212	3.726
Arginine	14.500	8.286	33.143	10.357	7.142	10.357
Hydroxyproline	27.292	5.955	23.818	11.909	5.133	11.909
Aspartic acid	13.793	6.650	24.106	8.728	4.299	8.728
Threonine	5.785	1.436	8.207	5.129	1.769	5.129
Serine	10.244	3.134	14.104	6.269	2.026	6.269
Glutamic acid	30.531	10.608	48.495	13.639	5.225	13.639
Proline	32.583	3.594	43.125	21.563	6.196	21.563
Glycine	70.724	36.694	90.726	44.355	21.202	44.355
Alanine	29.000	11.908	48.261	18.803	5.403	18.803
Cysteine	3.333	1.429	7.143	2.857	1.231	2.857
Valine	5.408	0.975	3.900	1.950	0.924	1.950
Methionine	0.876	1.014	0.507	1.014	0.699	1.014
Isoleucine	3.388	2.113	5.634	2.817	1.214	2.817
Leucine	6.550	4.010	10.694	4.010	1.728	4.010
Tyrosine	1.885	2.105	4.209	2.105	0.907	2.105
Phenylalanine	5.030	2.324	5.810	2.905	1.502	2.905
Total	279.510	116.322	399.898	175.652	83.096	175.652

**Table 6 polymers-14-04749-t006:** The Hyp/Hyl molar ratio of extracted collagens.

**Amino Acid**	**Amount, mg**					
	**I**	**II**	**IV**	**V**	**VII**	**VIII**
Hydroxyproline	27.292	5.955	23.818	11.909	5.133	11.909
Hydroxylysine	3.214	1.350	3.780	1.350	1.746	1.350
	**Molarity, µM**					
Hydroxyproline	208.1	45.41	181.6	90.82	39.14	11.909
Hydroxylysine	19.82	8.324	23.31	8.324	10.77	8.324
Hyp/Hyl molar ratio	10.49	5.45	7.79	10.91	3.63	1.43

**Table 7 polymers-14-04749-t007:** Output of by-products of leather processing.

By-Product	Coarse Raw Materials of Cattle		Small Raw Materials of Cattle	
	%	kg/ton	%	kg/ton
fleshings	18.5–23.0	185–230	15–17	150–170
pelt trimmings	1.8–2.0	18–20	2–4	20–40
pelt split trimmings	13.5–14.0	135–140	–	–
pelt split	10–14	100–140	–	–

## Data Availability

Data will be provided after an embargo of 1 year.

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
