# Peer review of "Collagen Obtained from Leather Production Waste Provides Suitable Gels for Biomedical Applications"

_polymers, 2022, doi:10.3390/polym14214749_

Round 1
Reviewer 1 Report
The manuscript entitled “Collagen obtained from leather production waste provides suitable gels for biomedical applications " represents a Collagen obtained from non-tanned wastes from the Chinbar enterprise (Kyiv, Ukraine) during different stages of the leather production: limed pelt, delimed pelt and fleshings after acid hydrolysis with acetic acid. However, the presented article could be published after revision for the followings;
1. Abstract should be rewritten to explain the aspects of the content including the main aim, method, results, and conclusion. It is seemed to me it could be better presented
2. The introduction should be modified for better justification for the aim of the current study.
3. Advantages and limitations of the study should be clarified.
4. References should be updated with more recent references till 2022 as there are many old references from the 1950s.
Author Response
- Abstract should be rewritten to explain the aspects of the content including the main aim, method, results, and conclusion. It is seemed to me it could be better presented
The abstract has been rewritten to reflect better the structure and the results of the paper.
- The introduction should be modified for better justification for the aim of the current study.
The introduction has been streamlined and several lines (throughout the document) have been rewritten to improve the English.
- Advantages and limitations of the study should be clarified.
A paragraph (3.7) has been added to clarify several limitations to be investigated after this work.
- References should be updated with more recent references till 2022 as there are many old references from the 1950s.
Reference 8 (originally, now 9), Bell E., Ivarsson B., Merrill C. (1979). Proc. Nat. Acad. Sci. USA, 76, 1274-1278. was replaced by Ehrlich, H. P.; Gabbiani, G.; Meda, P. Cell coupling modulates the contraction of fibroblast‐populated collagen lattices. J. Cell. Physiol. 2000, 184(1), 86-92.
Reference 14 (originally, now 15) is the one from 1950 but refers to the original Kjeldahl method, and an extra source (now ref 16) was already cited alongside said reference. We agree that references should reflect the state of the art, even for methodologies, but in this case, appropriate references were provided. We leave it to the Editor to decide whether one of the references should be scrapped.
Reference 16 (originally, now 17), Nowotny A. (1979) Protein Determination by the Biuret Method...was replaced with Zhou, P.; Regenstein, J. M. Determination of total protein content in gelatin solutions with the Lowry or Biuret assay. J. Food Sci. 2006, 71(8), C474-C479.
Reviewer 2 Report
The author Miastreko et al. investigated the collagen from leather manufacturing at different stages. The author also investigated its performance and compatibility with the HEK293 cell. It is intriguing that leather products cost so much waste and the idea of upcycling it is attractive. As collagen is a highly sought-after resource, in the realm of biomedical, cosmetic, and pharmaceutical industries. The experiment seems to be carried out correctly. This work is certainly interesting and well within the scope of MDPI polymers. There are a few comments I would like to raise and address for revision before the work can be considered for publication.
1. For your mineral content analysis, could the muffle furnace cost the oxidation and offset your result? Normally we use nitrogen or other inert gas for analysis.
2. I would also recommend the author do a quick intro on why the HEK293 cell was used for less experienced readers. For example, it’s a common benchmark for biotesting…etc.
3. In line 45, there is a larger gap between the word “conserved and “among”. There might be an extra space there.
Author Response
- For your mineral content analysis, could the muffle furnace cost the oxidation and offset your result? Normally we use nitrogen or other inert gas for analysis.
We understand the question of the reviewer. However, the mineral content analysis was performed according to ISO 4047:1977 | IULTCS/IUC7 for leather industry analysis. This has been added to paragraph 2.5.
- I would also recommend the author do a quick intro on why the HEK293 cell was used for less experienced readers. For example, it’s a common benchmark for biotesting…etc.
This sentence has been added in paragraph 3.6:
This cell line is among the most frequently used mammalian cell lines for a wide range of applications including recombinant protein production, manufacturing of viral vectors, cancer research, cytotoxicity and biocompatibility studies [25,26].
as well as these references
Zoia, L.; Binda, A.; Cipolla, L.; Rivolta, I.; La Ferla, B. Binary Biocompatible CNC–Gelatine Hydrogel as 3D Scaffolds Suitable for Cell Culture Adhesion and Growth. Applied Nano 2021, 2, 118–127. https://doi.org/10.3390/applnano2020010.
Pulix, M.; Lukashchuk, V.; Smith, D.C.; Dickson, A.J. Molecular characterization of HEK293 cells as emerging versatile cell factories. Curr. Opin. Biotech. 2021, 71, 18–24. https://doi.org/10.1016/j.copbio.2021.05.001
- In line 45, there is a larger gap between the word “conserved and “among”. There might be an extra space there.
This has been corrected (and on a few other occasions as well).
Round 2
Reviewer 1 Report
The article could be accepted in the present form